# Safety–Security Analysis of Maritime Surveillance Systems in Critical Marine Areas

**Batu Şengül** [1][ID], **Fatih Yılmaz** [2][ID] **and Özkan Uğurlu** [3,*][ID]

1   Security Sciences Institute, Gendarmerie and Coast Guard Academy, Ankara 06830, Türkiye;
    bsengul@sg.gov.tr
2   Republic of Türkiye Ministry of Transport and Infrastructure, Ankara 06490, Türkiye;
    fatih.yilmaz@uab.gov.tr or yilmazf58@gmail.com
3   Faculty of Marine Science, Ordu University, Ordu 52200, Türkiye
*   Correspondence: ougurlu@odu.edu.tr or ozkanugurlu24@hotmail.com

**Abstract:** In today's world, wherein more than 80% of world trade is carried out by maritime routes, the safety and security of the seas where this trade takes place is of vast importance for nations and the international community. For this reason, ensuring the sustainable safety and security of the seas has become an integral part of the mission of all maritime-related entities. Using big data extracted from the seas and maritime activities into meaningful information with artificial intelligence applications and developing applications that can be used in maritime surveillance will be of great importance for augmenting maritime safety and security. In this article, data sources which can be used by a maritime surveillance system based on big data and artificial intelligence technologies and which can be established around sensitive sea areas and critical coastal facilities, are identified and a model proposal using this maritime big data is put forward.

**Keywords:** big data; artificial intelligence; maritime surveillance; maritime security; sustainability

## 1. Introduction

The seas have diverse activities such as freight and passenger transportation, tourism, research, production, and military issues. Ensuring the sustainability, safety, and security of these maritime domains represents a shared objective among stakeholders operating at both national and international tiers [1]. Between 2019 and 2022 alone, 780 maritime accidents, along with 604 recorded incidents of maritime banditry and armed robbery, occurred. In the initial half of 2023, 5 maritime accidents, 4 very serious, and 65 cases of maritime banditry and armed robbery against vessels were reported [2,3]. In a global context where over 80% of international trade is facilitated via maritime transportation [4], in order to ensure that maritime activities are carried out safely and securely [5], authorities need to possess a comprehensive understanding of maritime activities to be able to detect irregularities at sea and to establish efficient maritime surveillance mechanisms. Maritime surveillance can be defined as the ongoing and systematic monitoring of maritime domains in order to create a high level of situational awareness [6]. Maritime surveillance activities provide substantial benefits in detecting irregularities that may be encountered in various issues such as safety of navigation, protection of the marine environment, combating illegal fishing, combating irregular migration, preventing all kinds of smuggling activities, and ensuring overall security [7].

Technological systems employed for the sustainability and effective execution of maritime surveillance activities are commonly referred to as maritime surveillance systems. Today, maritime surveillance is conducted through a diverse array of data-generating sensors, including "vessel monitoring systems (VMS)", "automatic identification system (AIS)", ship and coastal radars, air and space "synthetic aperture radars (SAR)", optical systems in ports and coasts, and "vessel traffic services (VTS)". In maritime surveillance

activities, these sensors can be integrated [8]. The main purpose of maritime surveillance systems is to ensure that maritime activities are carried out according to the law as a result of extracting the necessary information from maritime data and to provide full situational awareness at sea by promptly identifying any maritime irregularities that may arise.

Data sources for sustainable maritime surveillance can be categorized into three distinct groups: sensors, predefined databases, and publicly accessible internet sources [9]. Nonetheless, conducting maritime surveillance using these data sources presents several challenges, including the technical capabilities of the data sources, the vastness of the maritime areas to be monitored, the variability of the image on the sea surface owing to factors such as waves, surface currents and wake, weather events such as precipitation and cloud density, limited visibility at night and in foggy weather, traffic density of the area to be monitored, and detection of diverse vessel types and sizes [10]. In order to discern maritime irregularity with the aid of artificial intelligence, the normal situation must first be fully comprehended and modelled in a way that can be used in artificial intelligence applications. According to Martineau and Roy [11], using technology to detect maritime irregularities has multiple purposes. These are the efficient use of manpower, the development of decision support systems, advance notification and prediction, and the creation a holistic and continuous marine picture. Accordingly, the first step in detecting maritime irregularities is analysing generally accepted and historical data. The second step is to compare and evaluate the past data with the data on the events that are currently taking place. In this way, deviations from anticipated behaviour, such as maritime irregularities, can be detected. The deployment of big data technologies to the maritime sector, much like in other domains, offers substantial opportunities.

## 2. Literature Review

The concept of big data first emerged in the late 1990s. With the rapidly developing technology, digitalization, and widespread use of the internet in the last two decades, the high potential of big data has been recognized by researchers, governments, and companies, and it has been incorporated into daily life and started to be used in numerous fields [12,13]. It is not possible to use classical data analysis methods in big data applications due to the fact that the data to be studied are very large in size, complex and dispersed, encompass different types of data from diverse sources, etc. [12,14,15]. If big data do not make sense, they will not go beyond meaningless and raw data piles. In order to make sense of big data and use them in decision-making processes, it is necessary to analyse them through various criteria, methods, and algorithms and transform it into the meaningful information needed. Without proper interpretation, big data remain as meaningless, unprocessed data stacks [16].

Milicevic and Obradovic [17] conducted a comprehensive investigation into the collection and analysis of cruise data using big data applications. Their study elucidated that big data applications can significantly contribute to the formulation of novel strategies for optimizing logistics network planning and enhancing energy efficiency. Furthermore, their research posited that such advancements could yield advantages in terms of cost reduction, mitigating environmental pollution, and augmenting maritime safety. The study also highlighted drawbacks such as data quantity and quality challenges, the maritime sector's scarcity of specialized data scientists, various security and privacy issues, and the lack of legal regulations to govern data management.

In a study conducted by İşleyen, Uçar, and Balo [18] on the identification of irregular migrant boat crossings in the Aegean Sea, an algorithm based on simulated annealing was proposed to solve the problem and the algorithm was tested on various scenarios. Remarkably, it was observed that within the allocated solution time, the algorithm consistently detected all targets in the least possible time for all 50 different scenarios. The study's findings suggest that this methodology holds the potential to significantly enhance the rapid response to illicit activities at sea, thereby helping to avert fatalities associated with migration endeavours. In addition, it was posited that the methodology put forth

in the study possesses applicability in various other domains, such as the prevention of maritime accidents through its utilization in ship traffic control, search and rescue missions, identification of fish schools, placement of surveillance cameras and satellites, and the determination of unmanned aerial vehicle routes.

Filipiak et al. [19] compared the effectiveness of traditional and big data approaches in identifying maritime irregularities. Utilizing datasets that comprised AIS reports for all tankers globally in 2015 for the detection of maritime irregularities, the time required for computations when employing traditional MSSQL databases as opposed to Apache SPARK is compared. It is concluded that the big data approach outperformed the traditional method, achieving more favourable outcomes in detecting the recognized maritime picture (RMP) and identifying maritime irregularities.

Zissis, Xidias, and Lekkas [20] developed an artificial neural network (ANN) designed to forecast the future positions, speeds, and courses of vessels. They planned to anticipate the future behaviour of many vessels in a large maritime area by the diverse behavioural patterns exhibited by each vessel. As a dataset, the information of three passenger vessels with different routes that sailed regularly between Greek islands in 2013 was employed. In the study, they found a very large accuracy margin by analysing different ship movements and inferring from the large amount of data they obtained, especially in short-term (0–15 min) predictions. In the long term (75–150 min), it was observed that the ANN built in the study had difficulty in predicting the sudden course and speed changes (especially in port manoeuvres). They concluded that their ANN has significant potential in the scheduling of port operations, route planning of vessels, detection of maritime irregularities, and ultimately, increasing "maritime domain awareness" as a whole.

Several academic studies in the domain of maritime surveillance systems, leveraging big data and artificial intelligence technologies, have been conducted by various researchers. These studies encompass diverse data sources and methodologies:

- AIS Data-Based Studies
    - Stróżyna et al. [21]
    - Cepeda et al. [22]
    - Yitao, Lei, and Xin [23]
- Radar Data-Based Studies
    - Huang et al. [24]
    - Ilcev [25]
    - Ma et al. [26]
- Optical Systems-Based Studies
    - Pantazis [27]
    - Zardoua, Astito, and Boulaala [28]
    - Karabulut et al. [29]
- Data Fusion Studies
    - Bloisia et al. [10] discussed data fusion from AIS, radar, and optical systems.
    - Cubber et al. [30] studied data fusion from optical systems and acoustic sensors.
    - Zhao et al. [31] and Galdelli et al. [32] explored the fusion of SAR satellite imagery with AIS data.

In addition to academic studies, many countries have implemented new systems and projects using artificial intelligence and big data. In particular, important projects in this field have been implemented within the EU and the systems and projects examined in this study are presented in Table 1.

**Table 1.** EU maritime surveillance systems and projects reviewed.

| Project | Platform | Surveillance Range | Oversight Purpose/Activities | Sensors | Database | Executor |
|---|---|---|---|---|---|---|
| COPERNICUS | Satellite-Ground Observation (Land–Sea–Air) | EU Maritime Jurisdiction | Obtaining large amounts of global data by fusing data collected from Copernicus satellites, air and sea sensors, and ground stations, transforming this data into meaningful information and using it in the services needed | Satellite family (Satellite-1A/1B [SAR], Satellite-2A/2B [multispectral optical sensors], Satellite-3A/3B [medium resolution optical sensor and altimeter], Satellite-5P [atmospheric chemistry sensor], Earth observation data [S-AIS, VTS, VMS, LRIT, IFS]) | Surface meteorological and oceanographic data, landforms and ice formations, sea maps, sea surface temperature, chlorophyll-a and pollutant data | European Union |
| SCANMARIS | Software Family | Up to 200 Nautical Miles (nm) | Uninterrupted monitoring of EU maritime jurisdictions, making sense of large amounts of complex data from different sources, autonomously detecting anomalies in the sea with modelling and machine learning algorithms and notifying users | - | Data from satellite, radar, AIS, RDF, VTS sensors and Traffic2000, Lloyds, Paris MoU, ICCATT, TF2000, EQUASIS, TROCS, SATI databases within the EU | French National Research Center |
| I2C | Land–Sea (Ships–Naval Aircraft)–Air (Aircraft–Airship) | Up to 200 nm | Creating a new generation of innovative maritime border surveillance systems to monitor all vessel movements at sea in order to detect maritime anomalies/ suspicious events and to identify and report associated threats in advance | HFSWR and FMCW radars, AIS | Flag state information, meteorological data, intelligence database | Naval Group (French) |

**Table 1.** *Cont.*

| Project | Platform | Surveillance Range | Oversight Purpose/Activities | Sensors | Database | Executor |
|---------|----------|--------------------|-----------------------------|---------|----------|----------|
| COMPASS2020 | Land (Operation Center)–Sea (1 Patrol Ship, 1 Unmanned Underwater Vehicle)–Air (1 High-, 2 Medium-Low-Altitude Unmanned Air Vehicle) | Up to 200 nm | Demonstrating that the coordinated use of manned and unmanned technology and vehicles in air, sea, and submarine can achieve more successful results in information gathering and rapid response to maritime surveillance needs, increasing situational awareness at sea by providing cost-effective and reliable operational solutions to the coast guard and maritime authorities | Zephyr (radar and infrared camera) AR3 (electro-optical camera) AR5 (S-AIS, radar, electro-optical and infrared camera) | Naval picture transferred to the maritime operations centre | Portugal General Directorate of Maritime Enterprises |
| MARINE-EO | Software Family | EU Maritime Jurisdiction | Within the scope of Copernicus Security Service, to contribute to the development of EUROSUR legislation and CISE by creating an improved change detection system for monitoring anomalies around critical infrastructures and combating irregular migration, and strengthening international cooperation on maritime situational awareness | - | Copernicus System Sensors, CISE | Greek National Center for Scientific Research |
| EFFECTOR | Software Family | EU Maritime Jurisdiction | To create a data lake in order to detect different and new types of events that may be encountered at sea faster and more accurately, to improve maritime surveillance capabilities by applying data fusion and data analysis methods, to improve decision support, and to increase the interoperability of maritime stakeholders | - | CISE and EUROSUR integrated national and international databases | French Naval General Secretariat |

**Table 1.** *Cont.*

| Project | Platform | Surveillance Range | Oversight Purpose/Activities | Sensors | Database | Executor |
|---|---|---|---|---|---|---|
| SPYGLASS | Land–Sea (Buoy/Platform) | EU Maritime Jurisdiction | To develop a low-cost, stealthy, and environmentally friendly contact detection method by collecting the refracted and reflected signals of GNSS signals over contacts on the ground with passive radars deployed on land, sea, and air platforms | Passive bistatic radar (PRB) | Naval image database was created in the command centre | ASTER S.P.A. Ltd. (Italy) |
| SAFESHORE | Land (Mobile-Fixed Trailer) | Up to 1 nm | Developing an effective system to detect small unmanned air vehicles (UAVs) that can be flown from civilian ships when they cross the country's maritime border, creating an autonomous and mobile maritime surveillance system to detect low-altitude flying targets | Meteorological sensors lidar (3D/2D) short/long range (0–1800 m) thermal and electro-optical camera passive acoustic sensor, passive radio detection | Created target characteristic database | Royal Military Academy of Belgium |
| RANGER | Land | Up to 200 nm | To create a high-capacity and innovative surveillance platform by combining innovative radar technologies with state-of-the-art early-warning solutions to detect, track, recognize, and identify ships at greater distances than existing maritime surveillance systems to reduce the response times of operational units and to increase the response capacity of operational units | OTH radar PE-MIMO radar | Copernicus Meteorology<br><br>AIS, VTS, CISE | UNEX SOFTWARE Ltd. (United Kingdom) |
| PROMENADE | Software Family | EU Maritime Jurisdiction | To carry out maritime surveillance activities with maximum efficiency by using big data and artificial intelligence technologies to generate meaningful information from maritime big data to identify risky vessels before they enter EU maritime jurisdictions | - | VDES System Data Lake National databases | Greek Ministry of Maritime and Island Policy |

## 3. Methodology

The primary objective of this research is to put forth a model for a sustainable maritime surveillance system, utilizing big data and artificial intelligence, to safeguard critical coastal facilities and maritime zones. To accomplish this objective, potential data sources for the implementation of such a maritime surveillance system were identified with the help of an examination of relevant European Union systems and projects. Subsequently, the analytic hierarchy process (AHP) methodology was employed to ascertain the most appropriate and sustainable maritime surveillance options for enhancing the maritime security of a sensitive facility, exemplified by the Akkuyu Nuclear Power Plant (NPP). Ultimately, the sustainable maritime surveillance model proposal is presented in line with the AHP results.

A portion of the maritime surveillance activities falls under the scope of intelligence activities. Most research within this domain typically involves military and commercial confidentiality, rendering many confidential pieces of information, methodologies, projects, and applications inaccessible through open sources. Another constraint is associated with the fictional maritime surveillance model introduced in this study, necessitating the integration of diverse engineering disciplines for its implementation. In order to overcome this limitation to some extent, attempts are made to exemplify the general logic of the system and the algorithms that can be used in the proposed model through three fictitious scenarios.

Preliminary interviews and structured interviews were conducted within the framework of the AHP method to determine the most appropriate data source for the surveillance of critical coastal facilities and sensitive marine areas. During the establishment of the AHP model, the criteria were determined based on the literature review conducted within the scope of the study and preliminary interviews with experts. Sub-criteria were formed based on the main criteria. Alternatives were determined based on the maritime surveillance practices in the EU and Türkiye.

Previous studies have proposed various model recommendations utilizing data sources either individually or in pairs. However, a maritime surveillance system model based on big data and artificial intelligence has not been encountered that comprehensively utilizes all examined alternatives. Additionally, the proposed model has been supported with example scenarios. At this point, the aim is to unveil the significant potential embedded in the application phase of this hypothetical model.

The AHP method was Id by Thomas L. Saaty in 1977 to solve multi-criteria decision-making problems. In the AHP method, criteria and alternatives are created according to the purpose determined to solve a problem. Criteria are the characteristics required to solve the problem and generally include the considerations necessary to solve the problem and other important factors related to the problem. Alternatives are the different solution options that can be used to solve the problem [33]. The AHP method can be used to weight the criteria in a subjective manner, or it can be used to make a choice between decision alternatives [34]. In the AHP method, determining the relative importance of the criteria is an important step on the way to the solution. At the point of determining the relative importance of the criteria, the superiority of the criteria and sub-criteria, if any, to each other in a binary manner is evaluated, and in this evaluation, the opinions of experts in the field related to the problem are utilized. For this reason, the knowledge, experience, and competence of the experts whose opinions are consulted are important in the process of determining the degree of importance by comparing different criteria in terms of both quality and quantity and, thus, in the solution approach of the problem.

First, the criteria, then the sub-criteria, and, lastly, the paired comparisons of the alternatives are carried out by assigning values between one and nine, thus determining the degree of significance. Next, matrices containing the decision alternatives are created.

Since each criterion will be compared with its own significance value, the diagonals of these matrices have a value of one.

$$A = \begin{bmatrix} 1 & a_{12} & \cdots & a_{1n} \\ a_{21} = 1/a_{12} & 1 & \cdots & a_{2n} \\ \vdots & \vdots & \vdots & \vdots \\ a_{n1} = 1/a_{n1} & a_{n2} = 1/a_{n2} & \cdots & 1 \end{bmatrix} \tag{1}$$

A denotes the pairwise comparison matrix, n denotes the number of compared elements, i denotes the row, j denotes the column, aij denotes the significance or weight of the compared elements with respect to each other, aij denotes the importance of criterion i with respect to criterion j, and aji denotes the significance of criterion j with respect to criterion i. In the next step, the pairwise comparison matrix is normalized (2). "Eigenvector" calculation (3) and "Eigenvalue" calculation (4) are performed. Then, the maximum "Eigenvalue Size" is determined, respectively (5). "Consistency Index" is calculated (6). Finally, "Consistency Ratio" is calculated by utilizing the "Random Index" table, which is determined (7).

$$A' = a_{ij}' = \frac{a_{ij}}{\sum_{i=1}^{n} a_{ij}} \tag{2}$$

$$w_i = \frac{\sum_{i=1}^{n} a_{ij}'}{n} \tag{3}$$

$$w' = A_w = \begin{bmatrix} w_1' \\ w_2' \\ \vdots \\ w_n' \end{bmatrix} \tag{4}$$

$$\lambda_{max} = \frac{1}{n}\left(\frac{w_1'}{w_1} + \frac{w_2'}{w_2} + \cdots + \frac{w_n'}{w_n}\right) \tag{5}$$

$$CI = \frac{\lambda max - n}{n - 1} \tag{6}$$

$$\text{Consistency Ratio} = \frac{Consitency\ Index}{Random\ Index\ (RI)} \tag{7}$$

| n | 2 | 3 | 4 | 5 | 6 | 7 | 8 | 9 | 10 |
|---|---|---|---|---|---|---|---|---|---|
| RI | 0 | 0.52 | 0.89 | 1.11 | 1.25 | 1.35 | 1.40 | 1.45 | 1.49 |

The AHP hierarchical network used for identifying the most appropriate data sources for maritime surveillance of a critical coastal facility (Akkuyu NPP) in this study is shown at Figure 1.

### 3.1. Criteria

3.1.1. Data Characteristics Criterion

Data characterization is a frequently emphasized topic in the process of making sense of big data, which consists of data management and data analysis [14,35]. Big data have been characterized by various characteristics in the literature. Although there is no consensus on exactly how many different characteristics big data have [36–38], it is seen that five characteristics, namely volume, velocity, variety, accuracy, and value, are generally emphasized in existing scientific studies, and these are called the "5Vs of Big Data". Any characteristic of big data can be used in different academic studies for specific requirements and desired areas [14]. For example, a data scientist may be interested in big data volume and data variety [39], while a data engineer may focus more on characteristics such as data accuracy and data processing speed [40]. A business or organization manager may be

interested in the value of big data to extract the valuable information they need through data analysis [41]. The sub-criteria related to the "Data Characteristics" criterion were determined as in accordance with the purpose of the model:

- Data velocity;
- Data variety;
- Data veracity.

### 3.1.2. Maritime Surveillance Zones Criterion

In the maritime surveillance systems and projects reviewed in this study, one of the most important criteria is the size of the maritime surveillance area, also known as the coverage area. The size of the surveillance area may vary depending on factors such as surveillance method, characteristics of surveillance systems and sensors, density of maritime traffic, topography, meteorological conditions and natural phenomena, training and capability of the personnel responsible for surveillance, surveillance purpose, financial resources and costs, strategic importance of the region, etc. In interviews with experts, the range of the surveillance system and the size of the surveillance area came to the fore as expectations from a maritime surveillance system [42,43].

Whether the detection distance is adequate in terms of the time required to respond to threats varies according to the speed of the threat. As an example, a boat planning a terrorist attack on a critical coastal facility with a speed of 40 knots, will reach the coastal facility within 3 min from a distance of 2 nm. This time is extremely limited in terms of carrying out all detection, identification, reporting, and response operations. For this reason, it is considered that there is a need to identify maritime threats to the facility in advance. In light of this information, it has been assessed that it is necessary to establish a maritime surveillance area in three zones on the seaward side of a critical coastal facility. For this reason, the critical facility shall serve as the central reference point. The sub-criteria related to the "Maritime Surveillance Zones" criterion were determined as in accordance with the purpose of the model:

- Zone I: a circle with a radius of 2 nm towards the sea;
- Zone II: a circle with a radius of 2–12 nm (territorial water limit);
- Zone III: a circle with a radius of 12–30 nm (territorial sea border–international waters).

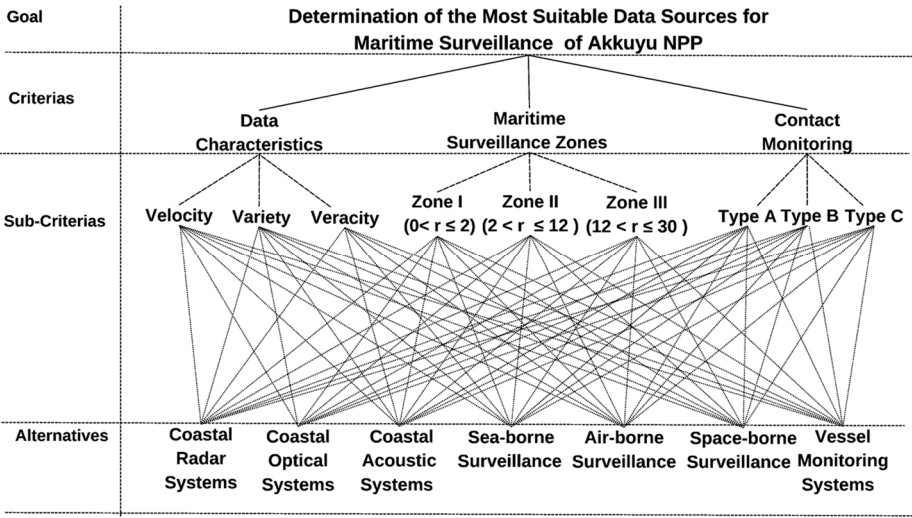

**Figure 1.** AHP hierarchical network for Akkuyu NPP.

### 3.1.3. Contact Tracing Capability Criterion

To ensure the effectiveness of maritime surveillance, it is imperative to successfully detect and identify all surface contacts within the designated maritime surveillance areas [44]. Within the scope of this study, military assets and weaponry associated with foreign states

are not considered as threats, and only civilian naval vessels that can be used for terrorist attacks and sabotage are taken into consideration. Accordingly, the sub-criteria related to this criterion are as follows:

- Type A: vessels mandated to possess AIS and LRIT devices in compliance with international regulations;
- Type B: vessels required to be equipped with AIS, fishing vessels monitoring system (FVMS), and vessel tracing module (VTM) devices in accordance with national regulations;
- Type C: private boats, small boats, sailboats, dinghies, rowboats, kayaks, canoes, pedal boats, Jet Skis, USVs, swimmers, etc., which are exempt from having any vessel tracking systems according to national and international regulations.

### 3.2. Alternatives

Diverse maritime surveillance systems are explained in detail in this chapter and the pros and the cons associated with these alternatives are summarized in Table 2.

**Table 2.** Comparison of maritime surveillance system alternatives.

| Alternative | System/Project Used | Pros | Cons |
|---|---|---|---|
| Coastal radar systems | I2C, Spyglass, Ranger | Widespread usage<br>High detection capability<br>Wide area surveillance<br>Continuous surveillance<br>Relatively low cost<br>Integrated operation | Affected by meteorological conditions<br>Risk of not being able to detect small, fast, and non-metallic boats and people on the water |
| Sea-borne surveillance | I2C, Spyglass | Possibility of continuous movement<br>Containing different sensors<br>High detection and diagnostic capability | Affected by meteorological conditions<br>Being widely manned and the risks of human error<br>High costs<br>Failure to perform the task without interruption |
| Coastal optical systems | SafeShore | Ability to provide high resolution images in day and night conditions<br>Continuous surveillance<br>High diagnostic capability<br>Integrated operation | Affected by meteorological conditions such as fog, haze, and precipitation<br>Diminishing effectiveness as distance increases |
| Air-borne surveillance | Compass2020, I2C | High mobility<br>Ability to scan large areas quickly and efficiently<br>Containing different sensors<br>High detection and diagnostic capability | Affected by meteorological conditions<br>Being widely manned and the possibility of human error<br>High costs<br>Failure to perform the task without interruption |
| Vessel monitoring systems | Copernicus, ScanMaris, Compass2020, MarineEO, Effector, Ranger, Promenade | Ability to present large amounts of diverse and accurate data<br>Integrated operation<br>Covering a large number of vessels by regulatory obligation | System shutdown/spoofing/malfunction<br>Need to be supported with other systems |
| Space-borne surveillance | Copernicus, CleanSeaNet, Effector, Promenade | Wide area surveillance<br>High resolution sea image transmission through SAR and optical sensors | High cost<br>Technical and technological limitations<br>Data communication and data rate problems |
| Coastal acoustic systems | SafeShore | Low cost<br>Ability to work in secrecy<br>Integrated operation | Ineffective in detecting overwater contacts<br>Effected by environmental conditions<br>Restricted range |

### 3.2.1. Coastal Radar Systems Alternative

Coastal radars have been used since the 1970s to remotely detect maritime contacts. Today, over the horizon (OTH) and high-frequency surface wave radars (HFSWR) are prominently used in maritime surveillance activities in the maritime jurisdictions of countries [24,25]. Each radar system possesses its own pros and cons. In general, radar systems excel in contact detection and tracking, especially at long distances, but have limited capacity in identification [45]. Maritime surveillance activities can also be carried out with passive radar systems using electromagnetic waves (GLONASS, GSM, LTE, etc.) available worldwide. It is seen that passive radar technologies, which have a place of use in the military field, especially thanks to their ability to be undetectable by enemy elements, have recently increased the idea that they can also be utilized for maritime and coastal surveillance applications [26]. Coastal radar systems are the primary data source underpinning the I2C, Spyglass, and Ranger projects.

### 3.2.2. Coastal Optical Systems Alternative

The optical sensors featured in this alternative encompass closed-circuit camera systems (CCTV), lidar, electro-optics, and thermal and high-resolution cameras. In their study, Kunz et al. [46] suggested that lidar systems are more effective in detecting small contacts (Jet Skis, small boats and boats, swimming people, etc.) on the water compared to radar systems and reported that they were able to detect buoys at a distance of approximately 5 nm in adverse sea and weather conditions and that this detection distance could be extended to approximately 11 nm. Today, the capabilities of optical sensors are evolving day by day with the technological advancements in this field. Although different parameters such as the characteristics of the contact, weather conditions, day/night conditions, and the location and altitude where the optical sensors are deployed significantly affect the contact detection and identification range, it is reported that optical systems that can provide images up to 20 nm are used in maritime surveillance activities [28,47]. In general, most optical surveillance system manufacturers specify in their product documentation that human detection up to 3 nm and vehicle detection up to 10 nm can be realized through the deployment of electro-optical and thermal cameras.

### 3.2.3. Coastal Acoustic Systems Alternative

In maritime surveillance activities, sonar systems and listening systems, including buoys and microphones deployed on shores, are used. Notably, research studies demonstrate that divers, unmanned undersea vehicles, and drones can be detected with these systems [30,48]. Although the range varies depending on the technical specifications of the acoustic sensors, the area to be listened to, the location where it is deployed, and the characteristics of the region, it is stated that surface contact detection can be extended up to 600 m with acoustic sensors [48,49]. The SafeShore project relies primarily on data collected from coastal acoustic systems as its principal information source.

### 3.2.4. Sea-Borne Surveillance Alternative

This maritime surveillance alternative includes both manned and unmanned CG vessels tasked with patrolling the seas. Many of these vessels come equipped with built-in radar, AIS, and optical systems. Surveillance activities carried out by maritime patrols are as significant today as they were in the past. Each sea asset can fulfil different requirements of maritime surveillance missions according to its technical capabilities [50]. Sea-borne surveillance offers distinct advantages over land-based systems in terms of mobility and contact identification, but disadvantages in terms of dependence on meteorological conditions, surveillance area width, and uninterrupted surveillance [51]. The primary data sources underpinning the I2C and Spyglass projects are sea-borne surveillance activities.

### 3.2.5. Air-Borne Surveillance Alternative

The airborne surveillance alternative comprises aeroplanes, helicopters, and UAVs. Airborne patrol and surveillance activities remain as important today as they were in the past [50]. In comparison to ground-based systems, airborne surveillance has advantages in terms of the diversity of data from different sensors, the ability to scan a wider area in a short time, high mobility, and close inspection and identification capability at the desired point and disadvantages in terms of cost, dependence on meteorological conditions, and the challenge of maintaining continuous surveillance capability [52,53].

### 3.2.6. Space-Borne Surveillance

The alternative to surveillance with space elements includes satellite systems. Depending on the intended use of the satellite, high-resolution optical systems and radar and communication systems can be found on the surveillance satellites. In space surveillance activities, several pivotal variables such as the number of satellites, orbital altitude, orbital speed, technical capabilities, and capabilities of the sensors on the satellite collectively influence the expanse of surveillance area and the resolution, detail, and transmission of images from the orbital assets to ground stations [54]. As a result of the examination of satellite surveillance systems, it is revealed that larger areas can be monitored thanks to this method compared to other surveillance alternatives, but the time to re-image the same area by satellites is in the range of approximately 2–3 days and the transmission of acquired satellite imagery to ground stations is not instantaneous, with an average delay of around 30 min [55,56]. It is a fact that surveillance activities with space assets have vast potential to be used much more widely in maritime surveillance activities, particularly given advancements in data transfer methods, the development of new technologies enabling the concurrent operation of micro-satellite groups for providing uninterrupted surveillance over a specific region, and the reduction in satellite technology costs. While these systems offer significant advantages, such as the ability to scan a very large area and transmit the sea picture with high resolution through SAR and optical sensors, they have disadvantages, such as cost, technical and technological constraints, data communication issues, and accuracy concerns [57]. The primary data source underpinning the Copernicus maritime surveillance system is derived from space assets.

### 3.2.7. Vessel Monitoring Systems Alternative

This alternative includes AIS and long-range identification and tracking of ships (LRIT), which are required to be available on vessels according to international regulations, and AIS, FVMS, VTM, etc., data sources, which are required to be available according to national regulations. AIS, in particular, serves as a primary data source for maritime surveillance activities [58,59]. The AIS system is devised for exchanging data over the VHF band, covering a 20–30 nm distance. However, since 2008, the coverage area has notably increased with the introduction of the S-AIS (Satellite AIS) system receiving AIS signals [60]. On the other hand, the LRIT system is a vessel tracking system introduced by the Maritime Safety Committee (MSC) of the IMO in 2006, based on the idea that a system should be established for the identification and tracking of vessels over extended distances, particularly within the scope of global counterterrorism measures [61]. Beyond these systems, many other applications are utilized individually by countries, such as fishing vessels, recreational vessels, vessels receiving discounted fuel, etc.

### 3.3. Expert Opinion

In the AHP method, expert opinion plays a crucial role in conducting pairwise comparisons between criteria and alternatives. In the context of this study, seven experts in the field of maritime safety and security were engaged. While selecting the experts, several criteria were sought to ensure a systematic and efficient data collection process:

- The experts should be professionals who are actively working in the Akkuyu NPP responsibility area.

- The experts should have experience in maritime safety and security of critical facilities in different regions.
- The experts should demonstrate knowledge in the field of electrical/electronics.
- Each interview session should last 120 min.

The experts were presented with a series of 201 pairwise comparison questions and tasked with scoring them between one and nine. AHP evaluations were conducted via online interviews with experts 1, 2, 4, 5, and 6 and face-to-face interviews with experts 3 and 7. Each interview session was completed within the planned 120 min duration. The key features of experts used in this study are listed in Table 3.

**Table 3.** Features of the experts.

| No. | Experience | Education | Graduation | Sector |
|---|---|---|---|---|
| 1 | 23 years | PhD | Maritime Transportation Management Engineering | Public Education |
| 2 | 23 years | PhD | Electrical and Electronics Engineering | Private Defence Industry |
| 3 [1] | 22 years | MSc | Electrical and Electronics Engineering | Public Maritime Safety |
| 4 | 19 years | MSc | Radar Specialization | Public Maritime Safety |
| 5 | 18 years | MSc | Electrical and Electronics Engineering | Public Maritime Safety |
| 6 [1] | 6 years | BSc | Maritime Transportation Management Engineering | Public Maritime Safety |
| 7 [1] | 3 years | BSc | Maritime Transportation Management Engineering | Public Maritime Safety |

[1] Involved in critical coastal facility maritime security.

## 4. Findings and Discussion

As a result of AHP, the priority importance levels of the criteria in providing maritime surveillance of a critical coastal facility were determined as

- 47% for "Data Characteristics";
- 29% for "Contact Monitoring Capability";
- 24% for "Size of Sea Area to be Monitored" (Table 4).

**Table 4.** Features of the experts.

| Criteria | Weight | Radar | Optics | Acoustics | Sea Assets | Air Assets | Space Assets | VMS |
|---|---|---|---|---|---|---|---|---|
| Data Characteristics | 0.469 | 0.083 | 0.077 | 0.031 | 0.106 | 0.095 | 0.034 | 0.045 |
| Data Velocity | 0.071 | 0.018 | 0.012 | 0.010 | 0.012 | 0.012 | 0.001 | 0.005 |
| Data Variety | 0.124 | 0.015 | 0.011 | 0.006 | 0.031 | 0.031 | 0.013 | 0.017 |
| Data Veracity | 0.274 | 0.050 | 0.054 | 0.014 | 0.063 | 0.052 | 0.019 | 0.022 |
| Size of Sea Area to be Monitored | 0.238 | 0.061 | 0.033 | 0.010 | 0.054 | 0.047 | 0.017 | 0.017 |
| Zone I | 0.069 | 0.015 | 0.017 | 0.005 | 0.016 | 0.011 | 0.002 | 0.003 |
| Zone II | 0.126 | 0.036 | 0.014 | 0.004 | 0.028 | 0.027 | 0.008 | 0.009 |
| Zone III | 0.043 | 0.010 | 0.002 | 0.001 | 0.009 | 0.009 | 0.006 | 0.005 |
| Contact Tracing Capability | 0.293 | 0.057 | 0.040 | 0.009 | 0.058 | 0.042 | 0.012 | 0.021 |
| Type A | 0.044 | 0.011 | 0.006 | 0.001 | 0.008 | 0.008 | 0.003 | 0.007 |
| Type B | 0.069 | 0.019 | 0.011 | 0.002 | 0.014 | 0.010 | 0.003 | 0.010 |
| Type C | 0.126 | 0.027 | 0.023 | 0.006 | 0.036 | 0.024 | 0.006 | 0.004 |

In the context of the AHP, "Data Characteristics" have been identified as the most crucial criterion for ensuring maritime surveillance at a critical coastal facility. The features of maritime big data are of paramount importance in the continuous tracking and detection of threats to ensure effective maritime surveillance. This criterion is considered as the

cornerstone of the data management process in the literature, particularly emphasizing the necessity of data possessing various attributes for achieving the intended purpose in artificial intelligence-based maritime surveillance applications and anomaly detection [14,62–66]. In the Copernicus maritime surveillance system and projects such as ScanMaris, Marine-EO, I2C, Compass2020, SafeShore, Effector, Ranger, and Promenade, data characteristics are frequently considered. The prominence of this criterion aligns with the existing literature and the examined projects.

### 4.1. Evaluation of Sub-Criteria

In the provision of maritime surveillance for a critical coastal facility, the "Data Veracity" emerges as the most crucial sub-criterion within the "Data Characteristics" criterion (Table 4). The accuracy of existing data in the process of converting obtained data into meaningful information ensures the maximum efficiency of the process by identifying threats and preventing misjudgements [14,63,64]. For instance, the Copernicus maritime surveillance system considers data obtained from space components accurately after combining it with ground-based surveillance systems through calibration, correction, and cross-referencing methods before presenting it to end-users.

In the "Size of Sea Area to be Monitored" criterion, "Zone II" is identified as the most significant area (Table 4). This zone, covering a range of 2–12 nm, needs to be effectively surveilled to provide sufficient intervention time for relevant units before contacts within this zone pose higher risks and enter the special security area. In a broader context, this conclusion does not fully align with the studies and projects under review. Maritime surveillance systems need to monitor vast maritime areas beyond inland and territorial waters [50,67,68]. In particular, the detection of high-speed threats from the farthest possible distance will be critically important for ensuring maritime security at these and similar critical facilities, as it will provide response units with the time required for intervention [69].

The most important contact type in"the "Contact Monitoring Capability" criterion is "Type C" (Table 4). "Type C" contacts are defined as small vessels, Jet Skis, swimmers, unmanned surface vehicles (USVs), and the like, which are not required to carry any VMS device under international and national regulations. While threats to maritime security from "Type A" and "Type B" contacts could be more substantial in terms of violence compared to "Type C" contacts, the probability of attacks, sabotage, terrorism, and other maritime security threats originating from "Type A" and "Type B" contacts is lower. Therefore, the effective monitoring of these contact types is of great significance.

### 4.2. Evaluation of Alternatives

The priority importance levels of the alternatives were determined as follows: 24% for "Coastal Radar Systems", 23% for "Sea-borne Surveillance", 17% for "Coastal Optical Systems", 16% for "Air-borne Surveillance", 10% for "VMS", 6% for "Space-borne Surveillance", and 5% for "Coastal Acoustic Systems" (Table 5).

**Table 5.** Prioritization and importance assessment of the alternatives.

| No. | Alternative | Percent (%) |
| --- | --- | --- |
| 1 | Coastal Radar Systems | 23.56 |
| 2 | Sea-borne Surveillance | 23.20 |
| 3 | Coastal Optical Systems | 17.04 |
| 4 | Air-borne Surveillance | 16.21 |
| 5 | VMS | 9.73 |
| 6 | Space-borne Surveillance | 5.63 |
| 7 | Coastal Acoustic Systems | 4.63 |

According to the AHP evaluation results, "Coastal Radar Systems" stand out as the most important alternative for sustainable maritime surveillance of a critical coastal facility

(Table 5). This result is in line with other studies in the literature [24,25,28]. Coastal radar systems are very important and widely used technologies for contact detection in maritime surveillance activities. However, in order to ensure complete maritime surveillance and sustainability, they need to be supported by other systems such as optical systems, VMS, communication systems, and maritime databases [7,70,71]. While the most important advantages of coastal radar systems are uninterrupted surveillance and precise detection capability, the disadvantages are that they are affected by meteorological conditions and there is a risk of being unable to detect small, fast, and non-metallic vessels and people on the water [72].

Conducting maritime surveillance activities with maritime assets (sea-borne surveillance) was identified as the second most important alternative (Table 5). The most important advantages of maritime assets are their continuous mobility and high diagnostic capability, while the disadvantages are the characteristics of the maritime assets, their technical capabilities, being affected by meteorological conditions, being widely manned nowadays and the risks of human errors, high costs, and difficulties in performing uninterrupted missions. In the coming period, with the increase in USV fleets and the advantages of USVs, the priority importance of sea-borne surveillance activities is likely to increase [73].

Coastal optical systems also stand out as an important alternative for the sustainability of maritime surveillance (Table 5). While the most important advantages of this alternative are its ability to provide high-resolution imagery in day and night conditions, as well as its uninterrupted surveillance and high diagnostic capabilities, its disadvantages are that it is affected by meteorological conditions such as fog, haze, and precipitation, and its effectiveness decreases with increasing distance [74]. The main data sources of the SafeShore project are coastal optical systems including lidar, electro-optics, and thermal cameras. The results obtained are in line with the reviewed literature [27,75]. Today, it is seen that the use of optical systems in maritime surveillance applications based on artificial intelligence has become widespread. More successful results are obtained daily in using data from optical systems for object detection and identification with image databases and machine learning modelling [66].

Air-borne surveillance has a similar importance value to coastal optical systems (Table 5). While high mobility and the ability to scan larger sea areas rapidly are the most important advantages, similar to maritime surveillance activities, the disadvantages include being affected by meteorological conditions, being widely manned and the possibility of human errors, high costs, uninterrupted execution of the mission, and short mission durations. Today, the importance of the use of UAVs in surveillance activities is increasing day by day and significant technological developments are taking place in this field. In the coming period, a significant portion of surveillance activities will be carried out by UAV fleets and the importance of this alternative is expected to increase [76]. UAVs with different types and technical specifications are important data sources in the Compass2020 project. Surveillance activities with air elements are also one of the main data sources of the I2C project.

VMS, which is at a very important point in providing maritime big data, appears less important in this study (Table 5), contrary to the literature [23,59,77]. The main reason for this situation is that this study mainly focuses on threats such as sabotage, terrorism, smuggling, illegal fishing, etc. Although suspicious contact is subject to VMS legislation, there is a possibility that these systems can be shut down and/or modified in illegal activities [78]. The result (Table 5) is understandable when the VMS is considered alone as a maritime surveillance system data source. However, it does not seem possible to consider a maritime surveillance system without VMS [28,58].

The surveillance with the space assets alternative (space-borne surveillance) is of relatively low importance value for the sustainability of the maritime surveillance system (Table 5). Surveillance with space assets is effective for long-term observation, strategic studies, and surveillance of large areas [79]. The low importance of this alternative is

understandable in the case of the need to provide tactical and operational level surveillance of a specific maritime area, as in this study.

The alternative of coastal acoustic systems is of low importance value (Table 2). Although effectively used to detect underwater contacts, existing sonar systems are technically inadequate for detecting surface contacts [30]. Even if it can detect contact, the limited detection range of microphones placed on the shore and the fact that they can be easily affected by ambient noise makes the coastal acoustic systems alternative an unreliable and ineffective option.

### 4.3. Maritime Surveillance System Model Proposal and Scenarios

The results obtained with the AHP application revealed the degree of importance of the data sources that can be used in a maritime surveillance system based on artificial intelligence. Accordingly, "Coastal Radar Systems", "Sea-borne Surveillance", "Coastal Optical Systems", and "Air-borne Surveillance" data sources have a share of 80% of the importance degree among all alternatives and, therefore, they are considered to be utilized as main data sources. The other alternatives, "VMS", "Space-borne Surveillance", and "Coastal Acoustic Systems" have a share of 20% in terms of importance and, therefore, can be utilized as auxiliary data sources. The model created according to the results obtained is presented in Figure 2.

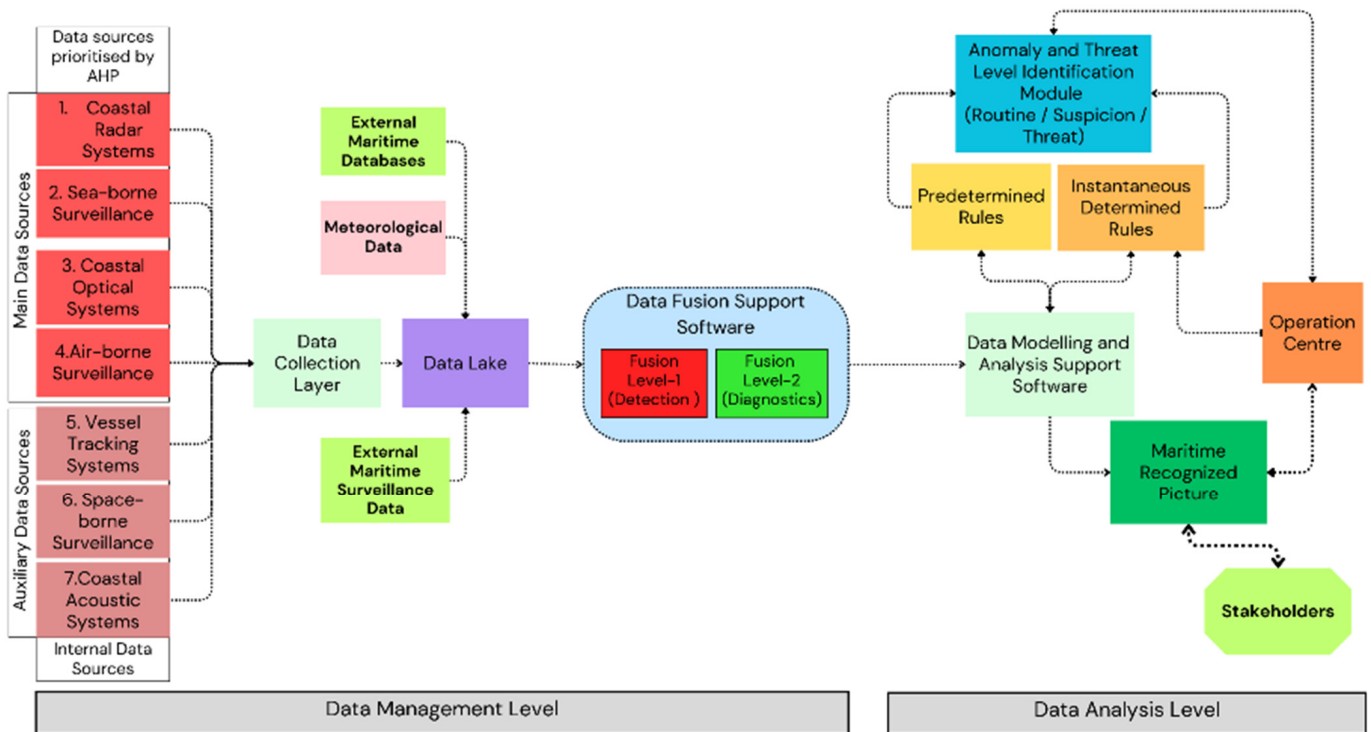

**Figure 2.** Proposed maritime surveillance system model.

The proposed model basically covers two processes. The first is the data management process and the second is the data analysis process. In the data management process, it aimed to collect internal source data from seven different data sources, four of which are main and three of which are auxiliary. Apart from the internal sources of the system, maritime databases, meteorological information, and data from other maritime surveillance system sensors are considered as external data sources.

The data collection layer transfers the data obtained from data sources to a central data lake. A data lake is a data storage and management mechanism frequently used in big data applications. Qualitatively and quantitatively, different types of data from different sources are collected here, processed through data fusion support software, and supported by software that includes detection and diagnosis stages. In this process, it aimed to transform

the data obtained from internal and external data sources into meaningful information and make them ready for the data analysis process. The data analysis process includes various methods used for big data management and analytics. These include data cleaning, integration, compression, pattern recognition, and predictive analytics techniques. The most important element of this process is data modelling and analysis support software. With this software, data modelling and analysis is performed using different methods such as big data analytics techniques, data exploration, data mining, and machine learning.

The data modelling and analysis utility outputs are used in two different modules. The first module is the defined sea image generation module. In this module, the defined sea picture generated as a result of analysing the observation data is displayed and data exchange is performed with the operation centre in a feedback manner. The aim is to identify all surface contacts within the region where surveillance is desired. The second module is the autonomous threat level determination module. This module aims to determine the threat level through predetermined and instantaneous security rules. Artificial intelligence techniques and machine learning algorithms classify the threat status as "routine", "suspicion", or "threat" by evaluating the organized and sorted data obtained from the data modelling and analysis utility software. The results are transmitted to the operations centre and feedback is received on a continuous basis. The security rules set in the system setup are included in the model as "predetermined security rules". These rules are defined in such a way that the operator cannot change them. In order to ensure that the system is flexible and user-friendly, new rules can be created by the operator at the tactical and operational level with "instantly determined security rules".

Scenarios

Scenario—1: The use of a 100 m long, 6000 deadweight chemical tanker with a cargo of jet fuel en route from the Western Mediterranean to the port of İskenderun in a terrorist attack on Akkuyu NPP.

While the vessel in question was 14 nautical miles off Gülnar/MERSİN, it changed its course approximately 90 degrees from east to north and proceeded to Akkuyu NPP at a speed of 12 kn. It did not respond to the calls made by the VTS over the radio. AIS device is operational and AIS data shows the port of arrival as İskenderun.

1. The coastal radar system and VMS automatically perform contact detection and identification operations as soon as the vessel enters Zone III. It starts tracking the vessel.
2. The coastal radar system detects the sudden course change of the vessel within the framework of predetermined rules. It compares and verifies this situation with AIS data. It checks whether the reason for the course change is due to the port of destination, vessel traffic, or meteorological conditions. (Predetermined security rule: categorize as suspicious any contact whose course changes by 30 degrees within the surveillance area without reasonable cause).
3. Compares the course and speed change, if any, with the port of arrival information in the AIS and maritime database and detects any deviation from the estimated time of arrival. (Predetermined security rule: Compare the AIS port of arrival and estimated time of arrival information with the port of arrival information in the maritime database. Categorize contacts that deviate from the standard route as suspicious).
4. The situation is reported to the operator on the electronic map in the operation centre with an orange code indicating that the vessel is suspicious and "Anomaly 1: Significant deviation from the course", "Anomaly 2: Contact's route is towards Akkuyu NPP ", "Anomaly 3: Course change not due to navigational hazard", "Anomaly 4: Course change not due to meteorological conditions".
5. The operator makes a radio call to the ship and receives no response. The operator calls the vessel via the phones registered in the maritime database and does not receive an answer. The VTS contacts the port authorities and CG units in the region. He learns that there is no force majeure notification that would require deviation from the route. The system marks the information that communication with the vessel could

not be established. The system reports the message "Anomaly 5: Unable to establish communication with surface contact". From this moment on, the ship is categorized as a threat and the ship is shown with code red on the electronic map. Relevant units are alerted and the response plan is activated.

Scenario—2: Terrorist attack to be carried out by a 22-m fishing boat departing from Erdemli/MERSİN fishing harbour for the purpose of fishing off Gülnar/MERSİN. (On the morning of the incident day, the operation centre receives unconfirmed intelligence that Akkuyu NPP may be attacked by a 150-m international freighter. The fishing vessel keeps its AIS, fishing vessels monitoring system, and VTM devices in working condition in order not to attract the attention of the CG units during its deployment. The fishing vessel arrives at a distance of 7 miles off Gülnar/MERSİN with the route and speed of routine fishery activity. The fishing vessel turns off its VMS devices here. It starts moving towards Akkuyu NPP at a speed of 8 kn).

1. Based on the intelligence received, the operator determines the instantaneous security rule that will enable cargo ships to be categorized as a threat if they enter the maritime surveillance area. (Instantaneous security rule 1: identify type A contacts entering Zone I as a threat. Instantaneous security rule 2: identify type A contacts entering Zone II as suspicious). Surveillance activities are started to be carried out with naval elements in the area.

2. The coastal radar system detects a contact entering the maritime surveillance area at a distance of 20 nautical miles. The contact is classified by the system in the routine category and with a yellow code and starts to be tracked. The VMS module checks for AIS, FVMS, and VTM data. The contact is identified and it is understood that it is a fishing vessel. VMS data and radar data are compared. No change is made by the system in the threat category since the data are compatible, the contact is identified, and it does not violate the previously and instantly determined security rules.

3. When the contact switches off the VMS devices, VMS data are no longer received and the system categorizes the contact as suspicious. (Predetermined security rule: classify vessels whose VMS data are interrupted/not received within the surveillance area as suspicious).

4. The situation is reported on the electronic map in the operation centre with an orange code indicating that the vessel is suspicious and the message "Anomaly 1: No data can be received via VMS".

5. It is detected by the coastal radar system and the patrol vessel operating in the area due to the intelligence received that the fishing vessel started to move towards Akkuyu NPP at a speed of 8 kn. The situation is automatically notified to the operator via the coastal radar system by the manned naval element by establishing communication with the operation centre. The system generates the message "Anomaly 2: Contact's route is towards Akkuyu NPP".

6. A radio call is made by the naval element to the ship and no response is received. The operator signals to the system that communication with the ship cannot be established. The system reports the message "Anomaly 3: Unable to establish communication with contact". The operator contacts the phone number of the ship owner registered in the maritime database and asks about the activity being performed. The operator realizes that the answers given are suspicious (the shipowner gives answers incompatible with the activity monitored regarding the activity of the fishing vessel) and makes suspicious/false information input to the system. The operator sees the message "Anomaly 4: Suspicious/false information given by the contact". From this moment on, the system categorizes the ship as a threat and the ship is shown with code red on the electronic map. Relevant units are alerted and the response plan is activated.

Scenario—3: Terrorist attack to be carried out by two 8 m long fibre-hulled inflatable boats moving very close to each other towards Akkuyu NPP at a speed of 40 kn from a distance of 40 nautical miles at 02:30 at night. (The boats will be separated from each other

at a distance of 2 nautical miles from Akkuyu NPP. One terrorist from the first boat will jump into the sea and infiltrate from the east coast of Akkuyu NPP with a surface vehicle called a "sea scooter", while the other boat will carry out a terrorist attack in the west of Akkuyu NPP).

1.  During the surveillance with air elements, two speedboats were detected by the patrol aircraft at a distance of 32 nautical miles to the south of Gülnar/MERSİN, rising at a speed of 40 kn towards the north and reported to the operation centre. This information is entered into the system by the operator and a new safety rule is defined. (Instantaneous security rule: categorize contacts with a speed of more than 25 kn entering Zone III as suspicious).

2.  The coastal radar system detects a high-speed contact at a distance of 9 nautical miles from Akkuyu NPP. The system checks whether the contact belongs to Coast Guard Command, Naval Forces Command, Turkish National Police, or any other public institution. The system categorizes the contact as a threat. (Predetermined security rule 1: categorize as a threat any non-public contact that shows a course towards Akkuyu NPP at a speed of more than 15 kn within the maritime surveillance area). An anomaly message is generated by the system ("Anomaly 1: Contact's route is towards Akkuyu NPP", "Anomaly 2: Contact which does not belong to a public institution with a speed of more than 15 kn within the surveillance area", "Anomaly 3: Contact exceeding 25 knots within surveillance area"). Subsequently, the response plan is put into practice.

3.  When the boats are 4 nautical miles from the shore, two fast boats are identified by the coastal optical system and the pursuit continues. It is determined that the boats are separated from each other. Necessary notifications are made to the response units by the operation centre.

4.  A swimmer is detected approaching the beach by optical systems monitoring the east coast of Akkuyu NPP and acoustic systems deployed on the coast. (Predetermined security rule 2: if a swimmer is detected by optical systems within Zone I, categorize as threat. Predetermined security rule 3: if an unidentified engine/propeller noise is detected by acoustic systems within Zone I, categorize as threat). Anomaly messages are generated ("Anomaly 4: Swimmer detection within Zone I", "Anomaly 5: Unidentified engine/propeller noise within Zone I") and new threat information is reported to response units.

The components of the proposed maritime surveillance system model according to scenarios are presented in Table 6.

**Table 6.** Scenarios—Maritime Surveillance Model Components.

| Scenarios | 1 | 2 | 3 |
|---|---|---|---|
| Data Sources | Coastal Radar System VMS | Coastal Radar System VMS | Air-borne Surveillance, Coastal Radar System, Coastal Optical System, Coastal Acoustical System |
| Fusion-1 (Detection) | Coastal Radar System | Coastal Radar System | Air-borne Surveillance, Coastal Radar System, Coastal Acoustical System |
| Fusion-2 (Diagnostics) | VMS | VMS Sea-borne Surveillance | Air-borne Surveillance, Coastal Optical System |

**Table 6.** *Cont.*

| Scenarios | 1 | 2 | 3 |
|---|---|---|---|
| Predetermined Security Rules | 1. Categorize as suspicious any contact whose course changes by 30 degrees within the surveillance area without reasonable cause<br>2. Compare the AIS port of arrival and estimated time of arrival information with the port of arrival information in the maritime database. Categorize contacts that deviate from the standard route as suspicious | 1.Classify vessels whose VMS data are interrupted/not received within the surveillance area as suspicious | 1. Categorize as a threat any non-public contact that shows a course towards Akkuyu NPP at a speed of more than 15 kn within the maritime surveillance area<br>2. If a swimmer is detected by optical systems within Zone I, categorize as threat<br>3. If an unidentified engine/propeller noise is detected by acoustic systems within Zone I, categorize as threat |
| Instantaneous Security Rules | - | 1. Identify type A contacts entering Zone I as a threat<br>2. Identify type A contacts entering Zone II as suspicious | 1. Categorize contacts with a speed of more than 25 kn entering Zone III as suspicious |
| Anomaly | 1. Significant deviation from the course<br>2. Contact's route is towards Akkuyu NPP<br>3. Course change not due to navigational hazard<br>4. Course change not due to meteorological conditions<br>5. Unable to establish communication with surface contact | 1. No data can be received via VMS<br>2. Contact's route is towards Akkuyu NPP<br>3. Unable to establish communication with contact<br>4. Suspicious/false information given by the contact | 1. Contact's route is towards Akkuyu NPP<br>2. Contact which does not belong to public institutions exceeding 15 knots within surveillance area<br>3. Contact exceeding 25 knots within surveillance area<br>4. Swimmer detection within Zone I<br>5. Unidentified engine/propeller noise within Zone I |

## 5. Conclusions

It has been concluded that maritime surveillance systems and projects based on artificial intelligence and big data have a significant potential to successfully realize sustainable maritime safety and maritime security, and they are being used by many countries today and will play an even more important role in the near future. In order to propose a maritime surveillance model based on big data and artificial intelligence, the study is based on a maritime surveillance system that can be realized around a sensitive sea area and/or a critical coastal facility. As a result, it has been observed that existing maritime surveillance systems can be improved and made more effective for the sustainability of surveillance activities in critical marine areas. Although there are studies on maritime surveillance systems based on big data and artificial intelligence technologies through VMS, radar systems, optical systems, air elements, naval elements, satellite technologies, acoustic systems, and maritime databases, the number of academic studies using these systems in an integrated manner is limited. This study will serve as an example for holistic studies to be carried out in the following period.

The proposed model can be used to provide sustainable maritime surveillance around any critical facility, island, fixed platform, or artificial island that can be built in the sea, sensitive maritime areas, maritime borders, and/or a region needing such as system at strategic, tactical, and operational levels, by making various modifications according to the needs.

**Author Contributions:** Conceptualization, B.Ş. and F.Y.; methodology, B.Ş. and F.Y.; validation, B.Ş.; formal analysis, B.Ş. and F.Y.; investigation, B.Ş.; writing—original draft preparation, B.Ş., F.Y. and Ö.U.; writing—review and editing, B.Ş., F.Y. and Ö.U.; supervision, F.Y. and Ö.U. All authors have read and agreed to the published version of the manuscript.

**Funding:** This research received no external funding.

**Institutional Review Board Statement:** Not applicable.

**Informed Consent Statement:** Not applicable.

**Data Availability Statement:** All data generated or analysed during this study are included in this published article.

**Conflicts of Interest:** The authors declare no conflict of interest.

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
