# Peer review of "Safety–Security Analysis of Maritime Surveillance Systems in Critical Marine Areas"

_sustainability, doi:10.3390/su152316381_

Round 1

Reviewer 1 Report

Comments and Suggestions for Authors

In this paper, the authors analyse methods and systems for maritime surveillance. This is completed with quite an extensive review of existing techniques, and finally, through the design of a model for defining in an analytic manner the criteria for defining the value of different techniques and the weight of different heterogeneous aspects used in maritime surveillance tasks.

In the end, they propose a model based on the analysis performed to identify the prominent criteria.

Even if the model looks quite complex to be implemented and integrated in a real situation, the study is interesting and, from a research point of view, of interest.

I would suggest using a unique terminology for the 3 Data characteristic terms (Data velocity, Data variety, Data veracity), while in Table 4, three other terms are used.

The presented scenarios are quite interesting; maybe it could be valuable, if admitted for length issue, to visually integrate them through a highlight of the components from the Figure 2 model, which are covered or connected by each respective scenario. I guess not all the components should be activated for all the scenarios.

Author Response

Reviewer #1:

General Comments to Reviewer 1:

Thank you very much for your useful comments and suggestions. We have made the necessary revisions to the study accordingly. We are of the opinion that the study is clearer as a result of these comments and suggestions being taken.

Comment 1 of Reviewer 1

I would suggest using a unique terminology for the 3 Data characteristic terms (Data velocity, Data variety, Data veracity), while in Table 4, three other terms are used.

Response to Reviewer 1, Comment 1

Thank you for your attention. Necessary revisions have been made in Table 4.

Comment 2 of Reviewer 1

The presented scenarios are quite interesting; maybe it could be valuable, if admitted for length issue, to visually integrate them through a highlight of the components from the Figure 2 model, which are covered or connected by each respective scenario. I guess not all the components should be activated for all the scenarios.

Response to Reviewer 1, Comment 2

The components of the proposed maritime surveillance system model according to scenarios are presented in Table 6. I hope we could respond to your revision request with this revision.

Reviewer 2 Report

Comments and Suggestions for Authors

Reviewing the presented materials, it is evident that the article focuses on critical aspects of maritime safety in the context of global trade, where over 80% of trade occurs through maritime routes. The authors underscore the significance of employing big data and artificial intelligence technologies for monitoring maritime activities to enhance safety. The introduction provides a broad overview of maritime activities, accompanied by alarming statistics on accidents and acts of piracy. The article extensively describes maritime surveillance systems, including technologies such as VMS, AIS, maritime and space radars, while highlighting challenges associated with utilizing diverse data sources. Despite some limitations in originality, the work stands as a valuable contribution to the field of maritime science, characterized by a high level of structure and clarity, and a solid grounding in current scientific research.

Author Response

Thank you for your positive comments on our article. We are very pleased that you accepted our article in its current form.

Reviewer 3 Report

Comments and Suggestions for Authors

This paper studies a sustainable maritime surveillance system model based on big data and artificial intelligence. The authors worked on analyzing criterions and alternatives for maritime surveillance systems and using the AHP methodology to select an appropriate system implementation. A few minor corrections that will improve the quality of the article are noted below. 

The order of citations to the references in the text should be reviewed. In general, the order of the reference list and the citation order of the references should be consistent.

The paragraph formatting of 3.3.1 needs to be corrected.

The headings of 3.2.1, 3.2.2 and 3.2.7 are the same and need to be revised.

Lines 524 to 549, paragraph formatting is wrong and needs to be revised.

How to ensure the  stability of the maritime surveillance system? The system proposed in this paper uses multiple data sources, if one of them fails, how to ensure the normal operation of the maritime surveillance system?

Comments on the Quality of English Language

Minor editing of English language required.

Author Response

General Comments to Reviewer 3:

First of all, we would like to thank you for your constructive comments and advice. We have tried to make the necessary revisions to the study by taking your comments and suggestions.  

Comment 1 of Reviewer 3

The order of citations to the references in the text should be reviewed. In general, the order of the reference list and the citation order of the references should be consistent.

Response to Reviewer 3, Comment 1

Thanks for your attention. Required revisions have been made.

Comment 2 of Reviewer 3

The paragraph formatting of 3.3.1 needs to be corrected.

Response to Reviewer 3, Comment 2

The paragraph formatting of 3.1.1 is corrected

Comment 3 of Reviewer 3

The headings of 3.2.1, 3.2.2 and 3.2.7 are the same and need to be revised

Response to Reviewer 3, Comment 3

The headings of 3.2.1, 3.2.2 and 3.2.7 are corrected and revised.

Comment 4 of Reviewer 3

Lines 524 to 549, paragraph formatting is wrong and needs to be revised.

Response to Reviewer 3, Comment 4

Lines 524 to 549, paragraph formatting is revised.

Comment 5 of Reviewer 3

How to ensure the  stability of the maritime surveillance system? The system proposed in this paper uses multiple data sources, if one of them fails, how to ensure the normal operation of the maritime surveillance system?

Response to Reviewer 3, Comment 5

You touched on a very good topic. The proposed model basically covers two processes. The first is the data management process and the second is the data analysis process. In the data management process, it is aimed to collect internal source data from 7 different data sources, 4 of which are main and 3 of which are auxiliary. Apart from the internal sources of the system, maritime databases, meteorological information and data from other maritime surveillance system sensors are considered as external data sources.

Comment 6 of Reviewer 3

Minor editing of English language required.

Response to Reviewer 3, Comment 6

Careful proof reading has been made. We are confident that this should be acceptable.

Reviewer 4 Report

Comments and Suggestions for Authors

Dear authors,

the work you have presented is very interesting. 

I have a few comments: 

1. Please indicate the purpose of the research. It has been written but please make it more explicit. Additionally, please indicate what is innovative in your research. 

2. Points 3.2.1, 3.2.2 and 3.2.7 have the same title. It seems that if they are separate points yo should have different titles.

3. The research indicates that the survey was conducted with 7 experts. Please indicate how many experts met the criteria and why these persons were chosen and not others.

4. There is an error in the citation. Please note publication 20. 

Author Response

General Comments to Reviewer 3:

Thank you very much for the constructive comments you have made to improve the quality of our study.  Please find the responses to the comments below:

Comment 1 of Reviewer 4

Please indicate the purpose of the research. It has been written but please make it more explicit. Additionally, please indicate what is innovative in your research. 

Response to Reviewer 4, Comment 1

In line with your comments, the following additional information has been added to the 4th paragraph of the methodology section. We think that the purpose of the article is more understandable after the revision. Additionally, the innovative findings of the article are explained in the same paragraph.

“Previous studies have proposed various model recommendations utilizing data sources either individually or in pairs. However, there hasn't been encountered a maritime surveillance system model based on big data and artificial intelligence that comprehensively utilizes all examined alternatives. Additionally, the proposed model has been supported with example scenarios. At this point, the aim is to unveil the significant potential embedded in the application phase of this hypothetical model”

Comment 2 of Reviewer 4

Points 3.2.1, 3.2.2 and 3.2.7 have the same title. It seems that if they are separate points yo should have different titles.

Response to Reviewer 4, Comment 2

The headings of 3.2.1, 3.2.2 and 3.2.7 are corrected and revised.

Comment 3 of Reviewer 4

The research indicates that the survey was conducted with 7 experts. Please indicate how many experts met the criteria and why these persons were chosen and not others.

Response to Reviewer 4, Comment 3

The experts consulted in this study were selected from individuals involved or currently engaged in maritime safety and security activities related to a critical infrastructure facility. Preference was given to experts knowledgeable in the electrical and electronics domain. All experts were chosen from those who have previously served or are currently serving at sea, and their details are outlined in section 3.3. Although a specific number of experts is not explicitly mentioned in the AHP method, literature commonly suggests selecting between 5 to 13 experts. The reason for choosing 7 experts is their possession of sufficient theoretical and practical background in the relevant field. Additionally, the limited number of experts available in this field, as commonly observed in most studies in the literature, also influenced the selection.

Comment 4 of Reviewer 4

There is an error in the citation. Please note publication 20. 

Response to Reviewer 4, Comment 4

Citation error ( number [20]) has been corrected.